# Mediation Analyses of the Role of Apathy on Motoric Cognitive Outcomes

**DOI:** 10.3390/ijerph19127376

**Published:** 2022-06-16

**Authors:** Mirnova E. Ceïde, Daniel Eguchi, Emmeline I. Ayers, David W. Lounsbury, Joe Verghese

**Affiliations:** 1Division of Cognitive and Motor Aging, Albert Einstein College of Medicine, Bronx, NY 10461, USA; emmeline.ayers@einsteinmed.edu (E.I.A.); joe.verghese@einsteinmed.edu (J.V.); 2Department of Psychiatry and Behavioral Sciences and Medicine, Montefiore Medical Center, Bronx, NY 10467, USA; 3Medical Program, Albert Einstein College of Medicine, Bronx, NY 10461, USA; deguchi@mail.einstein.yu.edu; 4Department of Epidemiology & Population Health, Albert Einstein College of Medicine, Bronx, NY 10461, USA; david.lounsbury@einsteinmed.edu

**Keywords:** apathy, depression, inflammation, gait, multimorbidity, motoric cognitive risk syndrome

## Abstract

Recent literature indicates that apathy is associated with poor cognitive and functional outcomes in older adults, including motoric cognitive risk syndrome (MCR), a predementia syndrome. However, the underlying biological pathway is unknown. The objectives of this study were to (1) examine the cross-sectional associations between inflammatory cytokines (Interleukin 6 (IL-6) and C-Reactive Protein (CRP)) and apathy and (2) explore the direct and indirect relationships of apathy and motoric cognitive outcomes as it relates to important cognitive risk factors. *N* = 347 older adults (≥65 years old) enrolled in the Central Control of Mobility in Aging Study (CCMA). Linear and logic regression models showed that IL-6, but not CRP was significantly associated with apathy adjusted for age, gender, and years of education (β = 0.037, 95% CI: 0.002–0.072, *p* = 0.04). Apathy was associated with a slower gait velocity (β = −14.45, 95% CI: −24.89–4.01, *p* = 0.01). Mediation analyses demonstrated that IL-6 modestly mediates the relationship between apathy and gait velocity, while apathy mediated the relationships between dysphoria and multimorbidity and gait velocity. Overall, our findings indicate that apathy may be an early predictor of motoric cognitive decline. Inflammation plays a modest role, but the underlying biology of apathy warrants further investigation.

## 1. Introduction

Current literature suggests that apathy is a predictor of dementia in older adults with preexisting neurological conditions such as mild cognitive impairment (MCI), stroke, or Parkinson’s disease (PD) [1,2]. Additionally, apathy has been associated with incident cognitive decline and dementia in community dwelling older adults [3,4,5]. Our group has also found an association between apathy and incident predementia syndromes including: non-amnestic mild cognitive impairment (MCI) and motoric cognitive risk syndrome (MCR), characterized by a slow gait and subjective cognitive complaints [6]. As the mechanistic pathway between apathy and dementia is unknown, clarifying the biological correlates of apathy will be important to develop novel treatment targets. A preponderance of studies has established a positive association between proinflammatory cytokines (including C-Reactive Protein (CRP), Interleukin-1 (IL-1), and Interleukin-6 (IL-6)) and depression, of which apathy is an important component [7,8,9,10,11,12]. Growing evidence suggests that apathy symptoms may mediate the previously described relationship between depression and cognitive decline [13,14]. Inflammatory stimuli reduce corticostriatal reward connectivity, ventral striatal neural responses to reward stimuli, dopamine in cerebrospinal fluid, cortical and subcortical gray matter volumes, and the integrity of white matter tracts within motivation- and reward-related circuits [15,16,17,18,19,20]. In particular, human studies of healthy volunteers acutely administered inflammatory stimuli and patients who received interferon for treatment demonstrated increased anhedonia and psychomotor slowing [18]. Given the impact of inflammation on neural reward pathways, we hypothesize that inflammation may play an important role between apathy and motoric-cognitive decline.

In addition to the association between apathy and cognitive decline, apathy is a feature of mild behavioral impairment (MBI), which is a concept that operationalizes late life neuropsychiatry symptoms as a “risk state” for dementia [21]. Thus far, β-amylase [22], neurofilament light [23], and cortical atrophy [24] have been correlated with MBI. Eurelings and her colleagues suggested that C-reactive protein levels (CRP) are associated with apathy in the preDIVA trial with older adults with cardiovascular risk [25]. Recently, deep white matter lesions mediated the relationship between CRP and apathy in community dwelling older adults [26]. However, these investigations have been limited to cohorts with high vascular burden and multimorbidity.

The primary objectives of this study were to (1) examine the correlational associations between inflammatory cytokines (Interleukin 6 (IL-6) and C-Reactive Protein (CRP)) and apathy and (2) explore the direct and indirect relationships of apathy and motoric cognitive outcomes (gait velocity, slow gait, and MCR) as it relates to important cognitive risk factors including dysphoria and multimorbidity in a healthy community dwelling cohort of older adults.

## 2. Materials and Methods

### 2.1. Study Population

347 out of 538 community dwelling older adults (age 65 and older) without dementia enrolled between 2013–2017 in the Central Control of Mobility in Aging study (CCMA) were included in this study. The primary aims of the CCMA cohort study are to determine the cognitive and neural predictors of mobility in late life. Study recruitment and procedures have been previously described in detail by Holtzer and colleagues [27]. Participants were recruited by mail and telephone from population lists in Westchester County, NY. Potential participants were screened for eligibility via a structured telephone interview to obtain verbal assent, assess medical history and mobility function [28]. Dementia was ruled out using the AD8 dementia screener [29], and the telephone-based memory impairment screen (MIS) [27,30,31,32,33]. A score > 1 on the AD8 or <5 on the MIS excludes the potential participant. Other exclusion criteria for enrollment into CCMA included the presence of dementia (previous physician diagnosis or diagnosed at baseline CCMA case conference), inability to walk with or without an assistive device, active neurological or psychiatric disorders severe enough to interfere with study assessments, presence of major visual or hearing loss, and recent or planned surgical procedures restricting walking. Participants who passed the telephone interview were enrolled and received comprehensive neuropsychological, psychological, and mobility assessments as well as a structured neurological examination. CCMA participants were followed longitudinally at yearly intervals. Baseline characteristics were collected during the clinical assessment, including age, gender, and years of education. General Health Status (GHS) (the total reported number of chronic medical conditions including depression, diabetes, hypertension, myocardial infarction, congestive heart failure, arthritis, stroke, Parkinson’s Disease, chronic obstructive pulmonary disease, and angina) was queried [34]. Between July 2013 and September 2014, a subset of participants were cross enrolled in a sub study to evaluate the biological correlates of fall [35]. Written informed consents were obtained at clinic visits according to study protocols and approved by the Albert Einstein College of Medicine Institutional Review Board.

### 2.2. Measures

Apathy. Currently there is no well-established gold standard instrument for assessing apathy [36]. Most studies have assessed apathy using subscales of depression assessment tools. Data reduction studies have consistently demonstrated an apathy-withdrawal as a distinct domain of the 30-item long form of the Geriatric Depression Scale (GDS) [37]. In order to distinguish apathy from dysphoria, we developed a novel measurement model to quantify apathy by applying confirmatory factor analysis (CFA) of the GDS [38], https://psychology-tools.com/geriatric-depression-scale/ (accessed on 1 June 2022) in a combined cohort of memory clinic patients and the CCMA cohort (*N* = 619). We identified a 2-factor solution utilizing 15 items of the GDS: dysphoria (eight GDS items: 6, 7, 8, 9, 10, 16, 17, and 25; *Cronbach’s α* = 0.75) and apathy (seven GDS items: 2, 4, 12, 19, 20, 21, and 28; *Cronbach’s α* = 0.63). Using the resultant measurement model, we computed scale scores for the pooled sample using the mean of the items in each factor. The dysphoria scale score was used as a covariate in the analysis.

Motoric cognitive outcomes. Gait velocity was assessed using the GAITRite system (CIR Systems, Havertown, PA, USA), a computerized walkway (dimensions 180 × 35.5 × 0.25 inches) with pressure sensors [39]. Participants are asked to walk on the mat at their ‘‘normal pace’’ for two trials without any attached monitoring devices. The GAITRite software automatically computes gait parameters, which includes gait velocity, based on footfall. Slow gait is a categorical variable that identifies participants with a measured gait velocity 1 SD below age and sex adjusted means with a prevalence of 15.1% [40]. MCR was operationalized similar to MCI and is defined as presence of cognitive complaints and slow gait in participants without dementia or mobility disability (inability to ambulate even with assistance or walking aids) [41].

Inflammatory Markers. While inflammation has been significantly associated with gait impairment in older adults [35], the biological underpinning of apathy and motoric cognitive outcomes, it is important to find a role for inflammatory pathways. Inflammatory biomarkers, such as Interleukin-6 (IL-6) and C-Reactive Protein (CRP), are suggested as predictors of apathy in the literature [25]. Serum levels of IL-6 and CRP were measured from frozen fasting blood samples. The samples were collected at a single wave between 2013 and 2014 as part of a sub study of the biological underpinnings of falls and are representative of extended periods [35]. The log of IL-6 and CRP were used in the analyses due to concerns about the normality.

### 2.3. Statistical Analysis

Bivariate analyses. We completed bivariate analyses of 347 community dwelling older adults enrolled in the Central Control of Mobility in Aging study comparing high and low apathy scale score as compared to the mean apathy scale score. Continuous variables were evaluated using independent samples *t*-test and categorical variables were evaluated by chi-square test and Fisher exact if greater than 20% of cells were less than 5. For this analysis, GHS was converted to a dichotomous multimorbidity index, in which 2 or more medical conditions was considered multimorbidity.

Mutivariable regression models. Linear regression models tested the hypothesized association of log IL-6 and log of CRP with apathy. We assessed the mediating effect of IL-6 on the association between apathy and MCR and slow gait, using logistic regression and linear regression models, respectively. Linear associations were reported using β-coefficients, while logistic associations were reported using odds ratio (OR) with 95% confidence intervals (CI), adjusting for age, gender, years of education, general health status and CFA-derived dysphoria scale score. All analyses were conducted in SPSS Version 28.0 (IBM Corp. Released 2021. IBM SPSS Statistics for Windows, Version 28.0, Armonk, NY, USA).

## 3. Results

Bivariate analyses. Table 1 highlights the pertinent baseline characteristics associated with higher levels of apathy. Older age, fewer years of education and multimorbidity (two or more chronic medical conditions) were all associated with a higher apathy scale score. Alternatively, vascular risk factors including hypertension, stroke, myocardial infarction, and congestive heart failure were not significantly associated with higher levels of apathy, in contrast to previous findings [42]. Additionally, behaviors, which increase vascular risk such as smoking and alcohol use, were not significantly associated with levels of apathy, and our findings agreed with previous literature [26]. Participants with higher apathy scale scores had significantly higher dysphoria scale scores and decreased gait velocity. Slow gait was borderline associated with higher apathy, while the predementia syndromes MCI and MCR were not. The inflammatory biomarker IL-6 was significantly associated with the higher levels of apathy while CRP was not.

Mutivariable regression models. We explored the association between inflammatory markers and apathy in linear regression models, which revealed that the log of IL-6 (0.04 95% CI: 0.01–0.08, *p*-value = 0.01) but not the log of CRP (−0.02 95% CI: −0.02–0.02, *p*-value = 0.80) was significantly associated with apathy, adjusted for age, gender, and years of education (Table 2). The association between IL-6 and apathy was no longer significant after adjusting for multimorbidity (general health status variable), a substantial confounder (25% change in the beta coefficient). Apathy was significantly associated with decreased gait velocity in the unadjusted model (β = −24.72, 95% CI: −34.88–−14.56, *p*-value = 0.01) (Table 3). While IL-6 was a modest confounder (14.5% change in the beta coefficient), the relationship remained significant after adjusting for IL-6, age, gender, years of education, multimorbidity, and dysphoria. Logistic regression models of slow gait and motoric cognitive risk syndrome were not significant.

Figure 1 shows a path diagram of the direct and indirect effects of variables of interest on gait velocity. Utilizing the product of the standardized coefficients, IL-6 was a modest mediator of the relationship between apathy and gait velocity (cm/s), while apathy substantially mediated the association between multimorbidity (number of comorbidities) and gait velocity and completely mediated the association between dysphoria and gait velocity.

## 4. Discussion

In this study, we investigated the direct and indirect associations of apathy with motoric cognitive outcomes as it related to important risk factors including inflammation and multimorbidity in a relatively healthy community dwelling cohort. Our bivariate analyses of the baseline characteristics revealed that dysphoria, IL-6, multimorbidity, and gait velocity were all significantly associated with higher levels of apathy, which we investigated further in regression models. Surprisingly, vascular diseases such as hypertension, myocardial infarction, stroke, and congestive heart failure were not significantly associated with levels of apathy, which differs from much of the previous literature on apathy in older adults [42]. Despite conducting our study in the CCMA cohort, a healthier cohort, multimorbidity was significantly related with the level of apathy [39]. In regression models, IL-6 but not CRP was associated with apathy. Unexpectedly, apathy was associated with gait velocity but not a slow gait or MCR. Utilizing a path diagram, we evaluated the mediation between these risk factors, apathy, and gait velocity and found that apathy mediated the previously described relationship between dysphoria and gait [43,44] as well as multimorbidity and gait [45]. While IL-6 was a modest mediator, it may be one component of a more complex relation between multimorbidity, apathy, and gait [46,47,48].

Our prior study of the CCMA cohort found that apathy was associated with incident MCR, but this evaluation did not find the same association [6]. Rather at a cross section, apathy was associated with a motor outcome (gait velocity), but not outcomes such as slow gait (one1 SD below the mean) or MCR, which might be more clinically relevant. Taken together, this suggests that apathy is an early risk factor for motoric cognitive decline, which precedes clinically notable motoric outcomes.

In terms of the neurobiology of apathy and inflammation biomarkers, our findings support at least a modest inflammatory correlation to apathy. In our CCMA cohort, we found that IL-6 level, but not CRP, was significantly associated with apathy, which differs from Eurelings’ previous findings that CRP was associated with apathy in the PreDIVA trial [25]. Yao’s study revealed that the mediating effects of deep white matter lesions in the relationship between CRP and apathy in community dwelling older adults [26]. The difference in cytokines (CRP vs. IL-6) could be explained by the health status of our cohort in contrast to other cohorts with a greater vascular burden. Additionally, it highlights a more complex biology than can be measured by two inflammatory cytokines.

While our findings elucidated some of the important predictors and mediators of gait velocity, we had some limitations. Firstly, our analysis was cross sectional, such that a clear pathway to motoric-cognitive decline cannot be assumed. However, our prior work demonstrated that apathy was associated with incident MCR and non-amnestic MCI, suggesting that apathy is an early predictor and mediator in the pathway [6]. Secondly, we only explored two cytokines (IL-6 and CRP), but there may be other cytokines or biomarkers that could be mediating the relationship between apathy and decreased gait speed. For instance, apathy has also been linked with other biomarkers associated with Alzheimer’s disease, including amyloid [49,50,51], tau [52,53,54], and brain-derived neurotrophic factor (BDNF) [55,56,57]. Given the mediating effects of apathy on the relationships between multimorbidity and dysphoria and gait, it is likely there are other biological correlates that are perpetuating motoric-cognitive decline. Thirdly, as this was a secondary analysis of an existing database, there was limited demographic information including the alcohol (one drink per week), which could bias any association between alcohol and apathy to the null. In addition, many participants were missing their BMI (height and weight), as this data was gathered at enrollment, but the biological underpinnings of the falls sub study was conducted at a later wave. Given the fluctuation of weight over time, it was not reasonable to try to impute the data.

Given our findings and the limitations of this study, our future analyses should include proteomic assays such as SomaScan in healthy cohorts [58]. This should generate deeper exploration into the inflammatory pathways but also promote investigation of additional important biomarkers associated with motor and cognitive function. Additionally, we plan to investigate how apathy may promote behaviors such as social isolation and decreased motor activity, which may promote inflammatory cytokines [59,60,61,62,63,64] and decrease brain-derived neurotrophic factor [65,66,67,68,69]. Another mechanism that warrants further exploration is the relationship between blood–brain barrier (BBB) disruption, which can occur in normal aging, and disease states including cerebrovascular disease and metabolic syndrome [70,71,72]. The loss of BBB integrity is associated with neuroinflammation and contributes to neurodegeneration [73,74,75].

## 5. Conclusions

Apathy is an important behavioral marker of underlying biological processes and an important early predictor of motoric-cognitive decline. Furthermore, the growing evidence [5,13,14,76] and our findings suggest that apathy may mediate previously described associations between depression, multimorbidity, and motoric-cognitive decline. Taken together, the assessment of apathy in older adults appears to be of clinical importance to identify high-risk populations early in the pathway of dementia-related pathogenesis. Apathy also represents a potential target for intervention in order to slow cognitive and functional decline.

## Figures and Tables

**Figure 1 ijerph-19-07376-f001:**
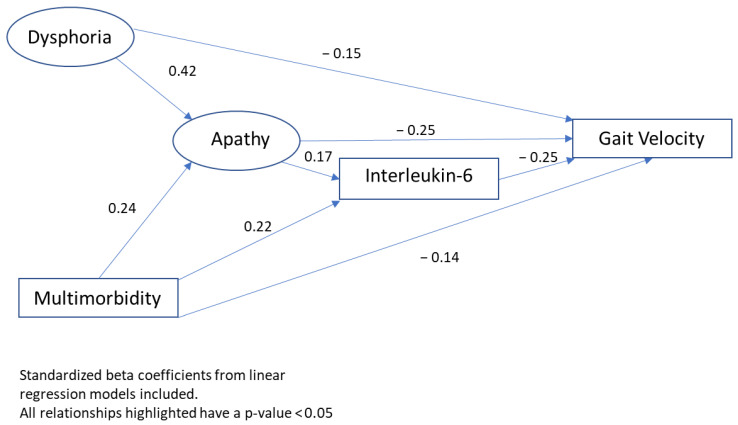
Path diagram of direct and indirect variable effects on gait velocity.

**Table 1 ijerph-19-07376-t001:** Baseline characteristics of apathy scale score.

Variable	Low *N* = 152 % (*N*)	High *N* = 193 % (*N*)	Statistic	*p*-Value *
**Age, mean (±SD), years**	76 (6.52)	77.5 (6.51)	T = −2.16	0.03
**Female**	40.8 (75)	59.2 (109)	X^2^ = 1.74	0.19
**Level of Education, mean (±SD), years**	15.3 (2.98)	14.3 (2.96)	T = 2.90	0.004
**Stroke**	0.7 (1)	2.6 (5)	X^2^ = 1.84	0.24
**Hypertension**	56.6 (86)	63.4 (121)	X^2^ = 1.62	0.22
**Diabetes**	17.1 (26)	19.7 (38)	X^2^ = 0.38	0.58
**Myocardial Infarction**	2.6 (4)	5.2 (10)	X^2^ = 1.42	0.28
**Congestive Heart Failure**	2.0 (3)	0 (0)	X^2^ = 3.82	0.09
**Multimorbidity Index ^a^**	42.8 (65)	61.1 (118)	X^2^ = 11.53	0.001
**Alcohol ^b^**	36.0 (54)	32.3 (62)	X^2^ = 0.52	0.49
**Smoking ^c^**	2.0 (3)	4.7 (9)	X^2^ = 1.80	0.24
**Dysphoria Scale Score, mean (±SD)**	0.04 (.10)	0.11 (0.16)	T = −4.68	<0.001
**Mild Cognitive Impairment**	12.5 (19)	18.7 (36)	X^2^ = 2.40	0.14
**Motoric Cognitive Risk**	7.2 (11)	10.9 (21)	X^2^ = 1.74	0.27
**Slow Gait (1 SD below the mean)**	12.5 (19)	20.8 (40)	X^2^ = 4.15	0.045
**Gait Velocity, mean (±SD), cm/s**	103.8 (21.54)	92.3 (22.76)	T = 4.77	<0.001
**logIL6, mean (±SD)**	−1.15 (152)	−0.95	T = −2.51	0.01
**logCRP, mean (±SD)**	0.06 (1.24)	0.10 (1.32)	T = −0.29	0.77

* *p* < 0.05 was considered significant. ^a^ Multimorbidity Index was defined as having 2 or more chronic medical conditions on the general health status scale. ^b^ Alcohol use defined as more than 1 alcoholic beverage per week. ^c^ Smoking use defined as currently or not smoking.

**Table 2 ijerph-19-07376-t002:** Linear regression of the association of inflammatory markers (logIL-6 and log hsCRP) and apathy *N* = 347.

Independent Variable	Log IL-6		Log hsCRP	
Model	Βeta (95% CI)	*p*-Value	Βeta (95% CI)	*p*-Value *
**Model 1**	0.05 (0.02–0.08)	0.001 *	0.00 (−0.02–0.02)	0.98
**Model 2**	0.04 (0.01–0.08)	0.01 *	0–0.02 (−0.02–0.02)	0.80
**Model 3**	0.03 (−0.004–0.06)	0.09	0–0.00 (−0.02–0.02)	0.72
**Model 4**	0.03 (−0.001–0.06)	0.06	0.00 (−0.02–0.02)	0.92

* *p*-value < 0.05. Model 1: unadjusted independent variable. Model 2: Model 1 + age + gender + years of education. Model 3: Model 2 + general health status (number of comorbidities). Model 4: Model 3 + Dysphoria Scale Score.

**Table 3 ijerph-19-07376-t003:** Linear regression of association between apathy and gait velocity *N* = 347.

Independent Variable	Apathy ^a^	
Model	Βeta (95% CI)	*p*-Value *
**Model 1**	−24.72 (−34.88–−14.56)	<0.001
**Model 2**	−21.13 (−31.22–−11.04)	<0.001
**Model 3**	−17.83 (−27.26–−8.39)	<0.001
**Model 4**	−16.57 (−26.19–−6.94)	0.00
**Model 5**	−14.34 (−24.84–−3.84)	0.01

* *p*-value < 0.05. Model 1: CFA derived apathy scale scores. ^a^ Apathy is measured using the confirmatory factor analysis derived scale score. Model 2: Model 1 + logIL6. Model 3: Model 2 + age + gender + years of education. Model 4: Model 3 + general health status (number of comorbidities). Model 5: Model 4 + CFA derived dysphoria.

## Data Availability

The data presented in this study are available on request from the corresponding author. The data are not publicly available due to privacy and HIPPA compliance.

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
