# Peer review of "Mediation Analyses of the Role of Apathy on Motoric Cognitive Outcomes"

_ijerph, 2022, doi:10.3390/ijerph19127376_

Round 1

Reviewer 1 Report

This is a very well written, cited and researched contribution describing the author’s efforts to determine “the underlying biological pathway” of that apathy associated with motoric cognitive risk syndrome. They have done numerous correlations/regressions to determine if the presence and degree of apathy, in an elderly population, is related to increased blood inflammatory cytokines and gait velocity.

What follows is in the spirit of improving an already fine contribution.

A general comment: Determining the underlying biological pathway for an emotion, as stated in the Abstract, seems rather ambitious; frankly, they don’t seem to have achieved this here. correlates of apathy” is more achievable and they have made a start with these findings. They have cited papers showing a relationship between inflammatory cytokines and depression to support their rationale for examining cytokines.  A general comment: That these are associated with dysphoria is not surprising to anyone who has observed individuals experiencing a COVID “cytokine storm,” immunotherapy, or other inflammatory medical conditions as these are often accompanied by depressed mood.

Lines 61-62 Is “cross-sectional analysis” … Correlational? Regression? Not clear what this is.

Lines 91-92. The questionnaire used to measure apathy appears to be accessible online: psychtools.info/gds/. If so, the authors may wish to include either the link or place the questionnaire in an Appendix as it is quite useful to the reader to see the actual questions used to evaluate apathy.

Line 92. “factory”…factor?

Lines 139-142. These sentences seem contradictory. What is the difference between “Slow Gait” and “decreased gait velocity”? Perhaps explained in lines 188-190? Perhaps this needs explanation earlier in “Methods.”

Discussion

The Discussion, nicely done, seems to cover the relevant issues related to this research.  Two questions come to mind: In the Introduction the authors allude to the longer-term goal of developing “novel treatment targets.” One wonders if anti-inflammatory medicines such as prednisone have beneficial effects for dysphoria and apathy and gait? Likewise, do anti-depressants improve gait velocity?

Reviewer 2 Report

The study purpose is a meaningful to show the usefulness for measurement of apathy especially in gerontology. However, there are some points that need to be revised.

1. This study set structured telephone interviews to rule out dementia. However, the references are nearly published before 20 years, and giving the impression that this is a somewhat outdated method. Please indicate why structural telephone interviews were used instead of other newer methods in this study.

2. Please specify the statistical software used in this study.

3. How defined "one drink per week" in terms of alcohol consumption?
Is it possible that the effects of alcohol consumption are not being measured on the Role of apathy.

4. Please explain why BMI (height and weight) was not measured in this study.

5. Do not draw arrows from right to left on the path diagram.

6. After revising the above points, please show the STROBE guidelines (https://www.strobe-statement.org/checklists/).

Reviewer 3 Report

Dear Authors,

I have read the manuscript entitled "Mediation Analyzes of the Role of Apathy on Motoric Cognitive Outcomes". The paper focuses on the direct and indirect associations of apathy and cognitive motor deficit in older adults. Based on numerous studies in the literature, the authors have given an important role to proinflammatory factors by studying the implications of Interleukin-6 (IL-6) and C-Reactive Protein (PCR) in 347 adults over the age of 65 enrolled in the Central Control of Mobility in Aging (CCMA) Study.

The authors of the study have a solid knowledge of motor and cognitive deficits, aspects presented in previous publications. The paper is presented in a clear and concise manner, respecting the requirements of the journal. The bibliographic references are in reasonable numbers, some of them being as recent studies as possible. The obtained results were presented in the 3 tables, which allows easy reading of statistical data.

However, I have a few comments:

1. As you mentioned in the Discussions section, this study has its limitations. There is no control group to check the relationship between serum IL-6 levels and apathy in healthy older adults. In this situation, is the statistical evaluation, correct? How do you justify this?

2. The presence of IL-6 as a proinflammatory factor has been reported in various studies with cognitive or cognitive-motor impairment. In the Introduction section, it is mentioned that inflammation can play an important role between apathy and cognitive-motor decline. Could you explain the mechanism by which IL-6, as a proinflammatory factor, contributes to the cognitive and motor decline?

3. The integrity of the blood-brain barrier and a number of neurotransmitters play an important role in neurodegenerative diseases. It would be advisable for the authors of the manuscript to provide some information in this regard in the Discussions section.

4. Please delete page 8/17 with lines 240 and 241.

Round 2

Reviewer 2 Report

The authors appropriately revised manuscript from review comments.